# The Role of Endoscopic Ultrasound in the Diagnosis of Gallbladder Lesions

**DOI:** 10.3390/diagnostics11101789

**Published:** 2021-09-28

**Authors:** Senju Hashimoto, Kazunori Nakaoka, Naoto Kawabe, Teiji Kuzuya, Kohei Funasaka, Mitsuo Nagasaka, Yoshihito Nakagawa, Ryoji Miyahara, Tomoyuki Shibata, Yoshiki Hirooka

**Affiliations:** Department of Gastroenterology and Hepatology, School of Medicine, Fujita Health University, 1-98, Dengakugakubo, Kutsukake-cho, Toyoake 470-1192, Aichi, Japan; hsenju@fujita-hu.ac.jp (S.H.); knakaoka@fujita-hu.ac.jp (K.N.); kawabe@fujita-hu.ac.jp (N.K.); teiji.kuzuya@fujita-hu.ac.jp (T.K.); k-funa@med.nagoya-u.ac.jp (K.F.); nmitsu@fujita-hu.ac.jp (M.N.); yo-hi@fujita-hu.ac.jp (Y.N.); ryoji.miyahara@fujita-hu.ac.jp (R.M.); shibat03@fujita-hu.ac.jp (T.S.)

**Keywords:** gallbladder carcinoma, endoscopic ultrasound (EUS), differential diagnosis, polypoid lesion, wall-thickening, staging of gallbladder carcinoma, contrast-enhanced EUS, EUS-guided fine-needle aspiration (EUS-FNA)

## Abstract

Gallbladder (GB) diseases represent various lesions including gallstones, cholesterol polyps, adenomyomatosis, and GB carcinoma. This review aims to summarize the role of endoscopic ultrasound (EUS) in the diagnosis of GB lesions. EUS provides high-resolution images that can improve the diagnosis of GB polypoid lesions, GB wall thickness, and GB carcinoma staging. Contrast-enhancing agents may be useful for the differential diagnosis of GB lesions, but the evidence of their effectiveness is still limited. Thus, further studies are required in this area to establish its usefulness. EUS combined with fine-needle aspiration has played an increasing role in providing a histological diagnosis of GB tumors in addition to GB wall thickness.

## 1. Introduction

Gallbladder (GB) diseases are relatively common and represent a variety of lesions including gallstones, cholesterol polyps, adenomyomatosis (ADM), and GB carcinoma. The most common disease is gallstones, affecting 10–15% of the adult population in the USA [1]. GB polyps have an estimated prevalence of approximately 5% in the global population [2,3,4], while GB carcinoma has an incidence of approximately 0.4 and 27 per 100,000 and in 100,000 people, respectively [5]. Northern India, Korea, Japan, and Central/Eastern Europe including Slovakia, Czech Republic, and Slovenia have also reported a higher prevalence than the worldwide average [6]. In contrast, GB carcinoma is rare in the western world (USA, UK, Canada, Australia, and New Zealand) [7].

Endoscopic ultrasound (EUS) has a high spatial resolution that can improve the diagnosis of GB polypoid lesions, GB wall thickness, and GB carcinoma staging. Vascularity can be evaluated using contrast-enhancing agents. Therefore, contrast-enhanced EUS may be useful for the differential diagnosis of GB lesions. However, the evidence of their effectiveness is still limited, and further studies are required in this area to establish its usefulness. EUS combined with fine-needle aspiration has played an increasing role in providing a histological diagnosis of GB tumors in addition to GB wall thickness. The role of EUS in the diagnosis of GB lesions has been demonstrated in many studies, and this review aims to summarize the role of EUS in the diagnosis of GB pathologies.

## 2. Literature Research

We conducted our literature research in the PubMed and Scopus electronic databases using combinations of the following key words: endoscopic ultrasound, gallbladder diseases, gallbladder neoplasm, and gallbladder carcinoma/cancer. We narrowed the research timeline to the period from 1991 to 2021. After initial selection based on exclusion criteria such as languages other than English, lack of access to a paper, and after having removed duplicates, we reviewed papers for further evaluation. From these papers, we mainly referred to our own papers and adopted 65 papers for this review. This review is written in the style of narrative review.

## 3. EUS Compared with Transabdominal Ultrasound to Detect GB Lesions

Transabdominal ultrasound (TUS) is used as a primary screening modality for gallbladder lesions because it is a relatively inexpensive, minimally invasive, and simple examination. However, EUS is considered superior to TUS for biliary system imaging because of the proximity of the duodenum to GB and higher resolution images by using higher ultrasound frequencies than TUS (5–12 versus 2–5 MHz) [8,9].

TUS has a sensitivity of 98% for gallstone detection [10]. However, TUS still has particularly some difficulty in identifying microlithiasis [10]. EUS can sometimes detect microlithiasis in the gallbladder in patients with grossly noncalculous biliary colic and normal TAUS findings. EUS demonstrated 92.6–100% and 55.6–91% of sensitivity and specificity, respectively, for the diagnosis of GB microlithiasis that had a negative result on TUS [11,12,13]. The benefit of using EUS over TUS has been best demonstrated in the diagnosis of microlithiasis.

The sensitivity of TUS in the detection of polypoid lesions of the gallbladder ranges from 36% to 99% depending on the presence of gallstones [14,15]. The current study found no studies for the detection of gallbladder polypoid lesions on EUS. The accuracy of TUS for the diagnosis of polypoid gallbladder lesions was reported to be 70–90% [16]. However, its diagnostic accuracy is strongly affected by the TUS technology and the ability of sonographers [17].

## 4. Differential Diagnosis of GB Lesions

GB lesions are broadly divided into protuberant lesions and wall-thickening lesions [18,19]. Protuberant lesions are defined as a focal elevation or a protrusion that can be distinguished from the surrounding mucosa [15,20]. Protuberant lesions are a comprehensive category that includes various diseases (Table 1). Protuberant lesions are first divided into two types (i.e., neoplastic and non-neoplastic protuberant lesions). Neoplastic protuberant lesions include adenomas and malignant lesions (e.g., GB carcinomas). However, non-neoplastic protuberant lesions include cholesterol polyps, inflammatory polyps, localized ADM, and hyperplasia. Neoplastic polypoid lesions should be treated by surgical resection, while non-neoplastic polypoid lesions can be observed serially.

Moreover, wall-thickening lesions denote lesions in which the GB wall is diffusely thickened. GB wall-thickening is defined as the GB wall measuring >4 mm and can be the result of various processes (Table 2) [21].

### 4.1. Differential Diagnosis of GB Protuberant Lesions

Several studies have evaluated EUS in the differential diagnosis of GB protuberant lesions [22,23,24,25,26,27,28,29]. The differential diagnosis for neoplastic and non-neoplastic lesions is based on size, number, morphology, surface contour, internal echotexture, and internal structure (Table 3). Among these findings, classifying them into pedunculated or sessile (broad-based) types is very important. Most pedunculated lesions are benign, and cholesterol polyps are the most common. Multiple polyps measuring ≤10 mm are highly likely to be cholesterol polyps [30]. However, malignant tumors are included in rare cases. These lesions are frequently incidental findings during abdominal examinations, and precisely distinguishing benign lesions from malignancies is important. EUS can visualize the layered structure of the GB and provide high-resolution images using high ultrasound frequencies. The characteristic findings of cholesterol polyps on EUS are a deeply notched granular surface and morular morphology. The internal echo is rough or granular, and highly echogenic punctiform foci reflecting cholesterolosis are visible (Figure 1) [25]. Peduncles are thin. Thus, they are often unobserved even on EUS [26]. Akatsu et al. [27] described the presence of hyperechoic spots, and multiple microcysts were important indicators of non-neoplastic lesions. Kimura et al. [24] described a granular contour, and a spotty internal echo pattern in the pedunculated polypoid lesions indicated benign pathology.

EUS shows an adenoma as a homogeneously isoechoic pedunculated or subpedunculated mass with a nodular or relatively smooth surface and an adenocarcinoma (pedunculated type) as a heterogeneously echogenic pedunculated mass with a nodular or smooth surface (Figure 2) [27,31]. Differentiation between adenomas and adenocarcinomas based on imaging is considered difficult. Thus, Cho et al. focused on relatively hypoechoic areas at the cores of the polyps, reporting the presence of such hypoechoic cores on EUS to be a strong predictive factor for neoplastic polyps. The overall accuracy of EUS in differentiating neoplastic from non-neoplastic lesions is 86.5–97% [22,25]. The accuracy of EUS in differentiating neoplastic from non-neoplastic polypoid lesions <10 mm was reported to be low [23]. Moreover, EUS is considered useful for guiding the treatment of larger pedunculated polyps [24].

EUS scoring systems have been proposed to differentiate between non-neoplastic and neoplastic GB protuberant lesions to aid diagnosis. Sadamoto et al. [32] reported the usefulness of an EUS score based on a coefficient of multivariate analysis: score = (maximum diameter in millimeter) + (internal echo pattern score; heterogeneous = 4, homogeneous = 0) + (hyperechoic spot score; present = −5, absence = 0). The sensitivity and specificity were 77.8% and 82.7%, respectively, in the differential diagnosis of neoplastic and non-neoplastic polyps using a cutoff score of >12. Choi et al. [33] have proposed another EUS scoring system for differential diagnosis of GB lesions between 5 and 15 mm based on layer structure, echo patterns, polyp margin, presence of a stalk, and the number of polyps. Moreover, the sensitivity and specificity were 81% and 86%, respectively, using a cutoff score of six.

Moreover, sessile lesions include localized ADM, carcinomas, and debris. Sessile GB carcinomas present with irregular internal echoes that are equal to or slightly hypoechoic to the liver parenchyma by EUS. Early GB carcinomas may be often accompanied by thickening of the inner hypoechoic layer around the main protuberant lesion [17,34].

EUS can visualize localized ADM as a sessile polypoid lesion with small cystic areas corresponding to the proliferation of Rokitansky–Aschoff sinuses (RAS; Figure 3) [27]. Comet tail artifacts are also occasionally observed owing to multipath reflection from RAS or intramural calculi. Several cases of GB carcinoma concomitant with ADM have been recently reported [35,36,37,38,39]. Therefore, the possibility of concomitant GB carcinoma with ADM in sessile lesions with multiple microcysts should be kept in mind.

### 4.2. Differential Diagnosis of GB Wall-Thickening Lesions

GB wall-thickening can be associated with a myriad of disorders. Therefore, GB wall-thickening poses difficulty in differentiating between benign processes (e.g., inflammation and malignant tumor). Thus, distinguishing early-stage cancer from benign wall-thickening of GB is important [40]. The contour of the lesion, patterns of wall thickness, intramural cystic space, and patterns of GB wall enhancement are used as differential points (Table 4). EUS can better define the characteristics of GB wall-thickening.

ADM can sometimes mimic GB carcinoma. The layers of a thickened GB wall are usually preserved in ADM, but there are microcysts with bright echoes arising from the cystic spaces. The thickened wall has a smooth surface but occasionally exhibits surface irregularity, reflecting hyperplastic changes. A key point in its diagnosis is the confirmation of the presence of cystic anechoic spots reflecting RAS inside the thickened wall (Figure 4) [26].

Xanthogranulomatous cholecystitis (XGC) is an uncommon disease involving chronic GB inflammation. Its clinical presentation is like that of cholecystitis, making it very difficult to distinguish from GB carcinoma because of marked tissue-destructive changes. In addition, imaging findings resemble those of GB carcinomas. EUS can sometimes visualize hyperechoic nodules in the GB wall, probably representing XGC. However, differentiation between benign and malignant types based on EUS alone is frequently difficult [41]. A report also exists that EUS-FNA was useful for preoperative differential diagnosis between GB carcinoma and XGC [42].

Epithelial height is increased, cellular proliferative activity is accelerated, and a mechanism from hyperplasia to dysplasia and carcinoma is speculated in hyperplasia of the gallbladder mucous membrane accompanying anomalous pancreaticobiliary junction because anomalous pancreaticobiliary junction permits reflux of pancreatic juice into the bile duct. EUS can delineate abnormal connections between pancreatobiliary ducts as clearly as endoscopic retrograde cholangiopancreatography [43]. GB lesions should be suspected to be malignant when EUS shows abnormal connections between pancreatobiliary ducts (Figure 5).

Differentiation from GB ADM and chronic cholecystitis is problematic in wall-thickening-type GB carcinoma. However, the mucous membrane is irregular or papillated, thickened areas do not have a uniform thickness, and the layered structure is ill-defined in GB carcinoma. Furthermore, microcysts and comet tail artifacts reflecting RAS are usually not observed. Mizuguchi et al. [44] reported that the loss of multiple layer patterns of the GB wall demonstrated by EUS was the most specific finding in diagnosing GB carcinoma. Kim et al. [45] noted that EUS findings of GB wall thickness exceeding 10 mm and hypoechoic internal echogenicity as independent predictive factors for neoplasm. However, differentiating malignant lesions from benign GB wall-thickening remains difficult.

## 5. GB Carcinoma Staging

EUS has been used in GB carcinoma staging because it can demonstrate the multilayer GB wall structure. In 1990, Mitake et al. [46] reported the effectiveness of EUS in determining tumor-invasion extent. The differentiation between early and advanced-stage tumors was 79.5% accurate, and the overall accuracy for depth of tumor invasion was 76.9%. No detailed examination was made in this study on whether each layer structure of the GB wall delineated by EUS corresponds to the mucosa, muscularis propria, subserosal layer, or serosa. In 1995, Fujita et al. [47] analyzed the layer structure of the GB wall delineated by EUS in detail using the pinning method for excised specimens. Consequently, the inner hypoechoic layer in a two-layer structure was clarified, and the hypoechoic middle layer in a three-layer structure includes not only the muscularis propria but also the fibrous layer of the subserosa. The outermost hyperechoic layer represents the adipose layer of the subserosa and the serosa. In 1998, Watanabe et al. [48] also analyzed the layer structure of the GB wall delineated by using intraductal ultrasonography, which has a higher frequency than EUS. The GB wall was composed of three layers: the innermost, middle, and outermost layers. Little fibrous tissue was seen and the second layer on sonograms was approximately identical to the muscularis propria in cases in which the thickness of the second layer was <500 µm. However, the second layer included not only the muscularis propria but also the fibrous layer of the subserosa in cases in which the thickness of the second layer was >500 µm. A consensus almost exists if GB wall-thickening is present, the inner hypoechoic layer includes not only the muscularis propria but also the fibrous layer of the subserosa, and the outer hyperechoic layer includes the adipose layer of the subserosa and serosa.

Fujita et al. [49] classified EUS images into four categories: type A, a pedunculated mass with a finely nodular surface and without abnormality of the neighboring gallbladder wall; type B, a broad-based mass with an irregular surface and no disruption of the outer hyperechoic layer of the gallbladder wall; type C, irregularity of the outer hyperechoic layer due to mass echo; and type D, disruption of the outer hyperechoic layer by mass echo (Figure 6). They then assigned the image types EUS to T stages for GB carcinomas. Type A would be a tumor in situ (Tis); type B, T1 or possibly T2; type C, T2; and type D, T3 or higher. Each of the four EUS image categories correlated well with the histologic invasion depth (Table 5).

The following is a summary of the diagnosis of the depth of invasion by EUS.

Those whose lesions can be diagnosed as pedunculated GB carcinomas can be diagnosed as Tis or T1a (M) depth of invasion.Sessile gallbladder carcinomas with thinning or irregularity of the outer hyperechoic layer can be diagnosed as gallbladder carcinoma with T2 (SS) invasion depth.In cases where the outer hyperechoic layer is retained, the depth of GB carcinoma invasion may extend to T1a(M), T1b(MP), or T2(SS), depending on the case, and differentiation is impossible even by EUS.

## 6. Contrast-Enhanced EUS

Contrast-enhanced EUS has been used to improve diagnostic accuracy based on the different levels of vascularity and blood flow that are found across different pathologic processes. Limited studies of contrast-enhanced EUS exist in the differential diagnosis of GB polyps or GB wall-thickening while many studies using contrast-enhanced EUS imaging have focused on pancreatic lesions. Hirooka et al. [50] reported that enhancement was observed in GB adenocarcinomas by contrast-enhanced endoscopic ultrasonography using sonicated albumin, but not in adenosquamous carcinomas and cholesterol polyps. They also reported that the depth of tumor invasion was assessed accurately in 11 of 14 cases (78.6%) in noncontrast EUS, while the assessment was accurate in 13 of 14 cases (92.9%) using contrast-enhanced EUS (Table 6). Latter studies were based on the second-generation contrast agents (e.g., SonoVue^®^ and Sonazoid^®^). The perfusion patterns were classified as diffuse enhancement, perfusion defect, and without enhancement in a contrast-enhanced harmonic EUS study by Choi et al. [51]. The vessels were categorized as regular spotty, irregular, or no vessels. This study reported that the presence of irregular vessel pattern and the perfusion defect on contrast-enhanced EUS can diagnose GB carcinomas in GB polyps measuring at least 10 mm with a sensitivity and specificity of 93.5% and 93.2%, respectively, versus 90.0% and 91.1% for conventional EUS. Kamata et al. [52] also reported that GB carcinoma was characterized by irregular vessels in the vascular image and heterogeneous enhancement in the perfusion image (Figure 7). The sensitivity, specificity, and accuracy for the diagnosis of carcinoma on contrast-enhanced harmonic EUS was 90%, 98%, and 96%, respectively, in this study.

Another study by Imazu et al. [53] using contrast-enhanced EUS in the differential diagnosis of GB wall-thickening demonstrated inhomogeneous enhancement as a strong predictive factor of malignant GB wall-thickening (Figure 8). The same study reported that overall sensitivity, specificity, and accuracy for diagnosing malignant GB wall-thickening for EUS and contrast-enhanced EUS, respectively, were 83.3% versus 89.6%, 65% versus 98% (*p* < 0.001), and 73.1% versus 94.4% (*p* < 0.001).

Xue Liang and Xiang Jing [56] reported a meta-analysis of contrast-enhanced ultrasound and contrast-enhanced harmonic EUS (CH-EUS) for the diagnosis of GB malignancy. The pooled sensitivities of CH-EUS and specificities were 0.92 and 0.89 (Table 6), respectively, in this meta-analysis. On CH-EUS, the heterogeneous enhancement could be indicative of malignant lesions with a sensitivity and specificity of 0.94 and 0.92, respectively.

However, further accumulation of knowledge is desired because no large-scale study on contrast-enhanced harmonic EUS in GB diseases to date.

## 7. EUS-FNA for GB Lesions

The EUS-FNA role in tissue sampling for pancreatic and gastrointestinal lesions has been established. However, its role in the diagnosis of GB lesions has not been elucidated. The pathological diagnosis of GB lesions to rely on cytological examination of the bile obtained by endoscopic transpapillary gallbladder drainage tube is often necessary [57]. However, cases are sometimes experienced where a definitive diagnosis cannot be made pathologically even with this method. In addition, cytology using endoscopic naso-GB drainage has problems (e.g., perforation of the cystic duct when using a guidewire). In such cases, the pathological search by EUS-FNA is useful if the tumor can be visualized by EUS. Hijioka et al. [58] have reported that FNA can be performed in GB lesions without compromising diagnostic performance or safety. Moreover, the diagnostic performance of EUS-FNA in GB lesions is high with a sensitivity, specificity, and diagnostic accuracy of 80–100%, 100%, and 83–100%, respectively (Table 7) [58,59,60,61,62,63], which was superior to endoscopic transpapillary GB aspiration with cytology [62] or endoscopic retrograde cholangiography-guided sampling [58]. No report exists of bile peritonitis or tumor seeding in EUS-FNA in GB disease.

EUS-FNA are not recommended for resectable GB carcinoma, because this procedure may induce biliary peritonitis and would also cause peritoneal dissemination, like transabdominal US-guided FNA cytology [65]. The strategy for obtaining tissue from gallbladder tumors is first to try to obtain tissue from gallbladder tumors by endoscopic retro grade cholangiography (ERC) biopsy, then from liver or lymph node metastases by EUS-FNA, and finally from gallbladder tumors by EUS-FNA [66]. Indications of EUS-FNA of gallbladder mass lesions should include the following:GB large lesions; only if the lesion can be punctured without the needle passing through the GB lumen (Figure 9) or lesions with the very thickened GB wall [67].In patients with gallbladder tumors accompanied by liver and/or lymph node metastasis, the liver and/or lymph node metastasis should be punctured before the gallbladder tumor is punctured.When it is difficult to categorize a lesion as benign or malignant, or when the surgery is extremely invasive, EUS-FNA should beconsidered.

When taking a sample from the gallbladder, it is important that the needle does not pass through the lumen of the gallbladder [68]. When directly puncturing the GB wall, it takes care to gain stroke distance by tangentially puncturing the gallbladder wall. However, the wall may move if the gallbladder lumen remains, and puncturing is often difficult. Thus, it is best to puncture the neck side of GB. In cases where lesions have invaded the liver, it is recommended to puncture either the liver parenchyma as the invasion site or the gallbladder wall that is in contact with the liver parenchyma. Regional lymphadenopathy is often noted in unresectable advanced GB carcinoma. EUS-FNA from the regional lymph nodes is preferable considering the risks (e.g., invasive biliary fistula and peritoneal dissemination).

## 8. Conclusions

EUS provides higher resolution images of GB lesions using the higher frequency and closer proximity to GB when compared to conventional TUS. The EUS is highly useful for the detection of GB lesions, the differential diagnosis of GB polypoid lesions and GB wall thickness, and the evaluation of staging of GB carcinomas. In the future, the importance of EUS in this field is expected to grow further with the use and establishment of contrast-enhanced EUS and EUS-FNA for GB lesions.

## Figures and Tables

**Figure 1 diagnostics-11-01789-f001:**
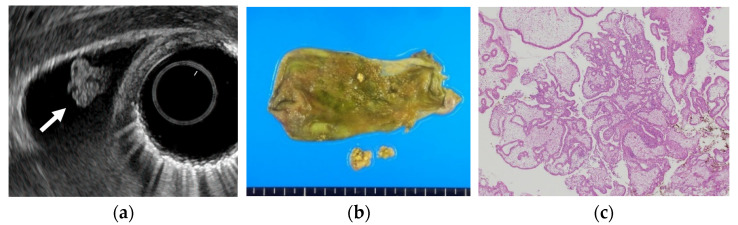
Cholesterol polyp. (**a**) EUS shows a cholesterol polyp as a granular-surfaced pedunculated lesion. The internal echo is heterogeneous with a hyperechoic spot (arrow). (**b**) Photograph of the gross pathologic specimen after cholecystectomy shows a yellowish polyp. (**c**) H-E stain of the specimen demonstrates an aggregation of foamy cells under the epithelium.

**Figure 2 diagnostics-11-01789-f002:**
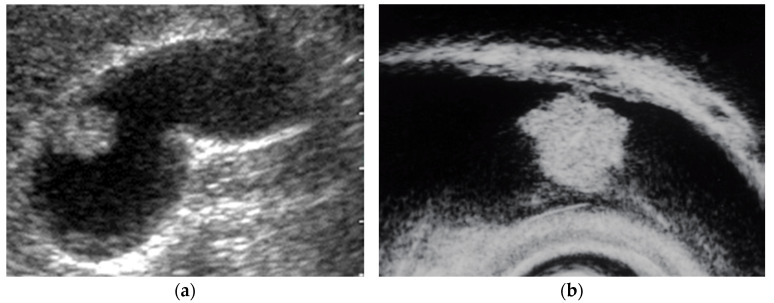
Pedunculated GB carcinoma. (**a**) TUS image shows a relatively smooth surface, solid internal echogenicity polyp, but TUS does not depict the nature of the base of the lesion. (**b**) EUS image shows a pedunculated lesion. This lesion was GB adenocarcinoma with invasion depth pT1a (M) as a result of surgery.

**Figure 3 diagnostics-11-01789-f003:**
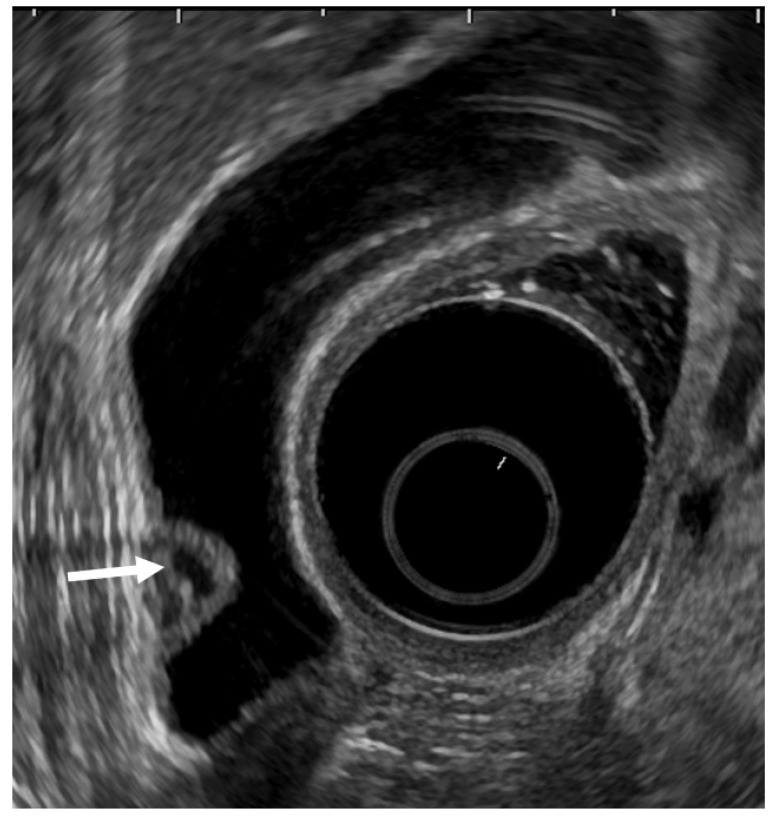
EUS image ADM (localized type). EUS shows localized ADM as a sessile polypoid lesion with anechoic areas (arrow) corresponding to RAS proliferation. The surface is relatively smooth.

**Figure 4 diagnostics-11-01789-f004:**
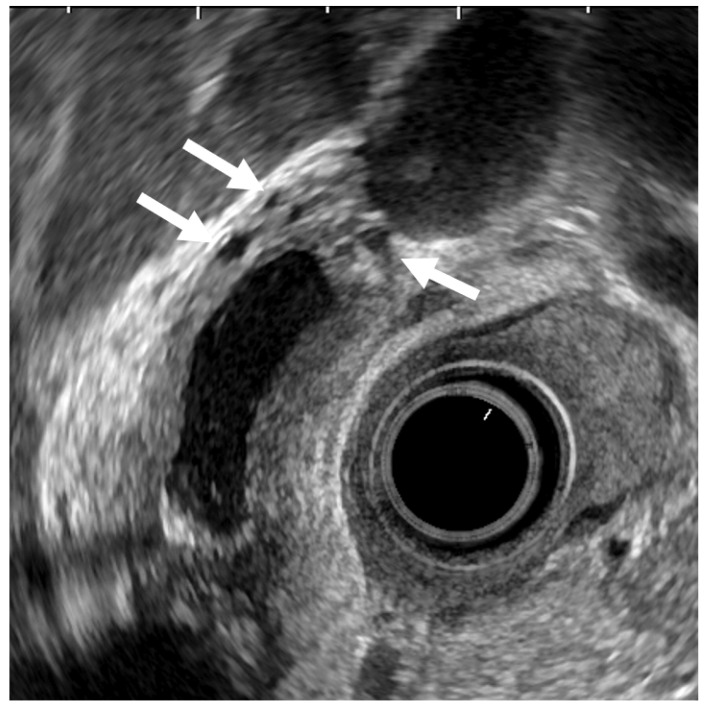
EUS image of ADM (diffuse type). The GB wall is diffusely thickened, and the layers of a thickened GB wall are preserved. Some anechoic areas (arrows) are visualized in the GB thickened wall.

**Figure 5 diagnostics-11-01789-f005:**
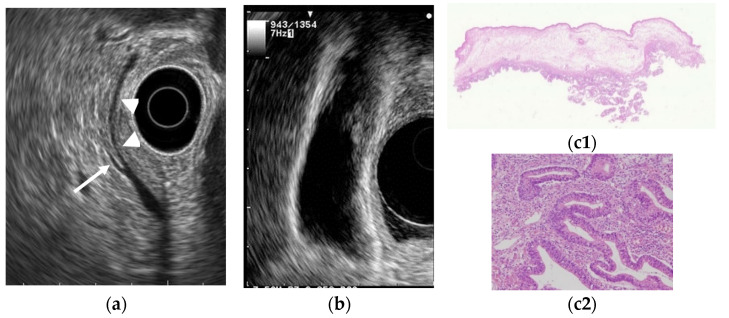
GB carcinoma is associated with pancreaticobiliary maljunction without biliary dilatation. (**a**) EUS shows the bile duct (arrowhead) and main pancreatic duct (arrow) communicated inside the pancreas. (**b**) EUS shows the irregular GB wall-thickening on fundus without wall layer structure disruption. (**c1**,**c2**) H–E stain of the specimen demonstrates adenocarcinoma with tumor in situ stage.

**Figure 6 diagnostics-11-01789-f006:**
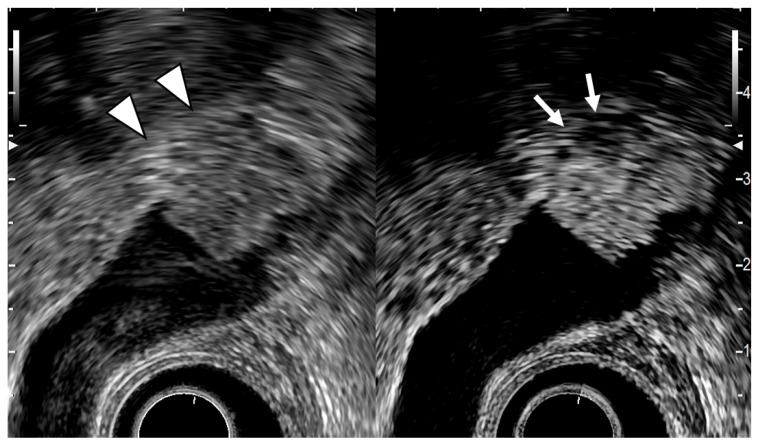
EUS image of GB carcinoma. The conventional EUS (**left**) shows that a broad-based elevated lesion is found at the fundus of the GB with hypoechoic (arrowhead) in the deep part of the lesion and rupture of the lateral hyperechoic layer. In the contrast-enhanced EUS (**right**), the contrast effect of most of the lesion is good, but the deep part of the lesion is poorly contrasted (arrow). It can be diagnosed from these findings as GB carcinoma with invasion depth T3a (SE).

**Figure 7 diagnostics-11-01789-f007:**
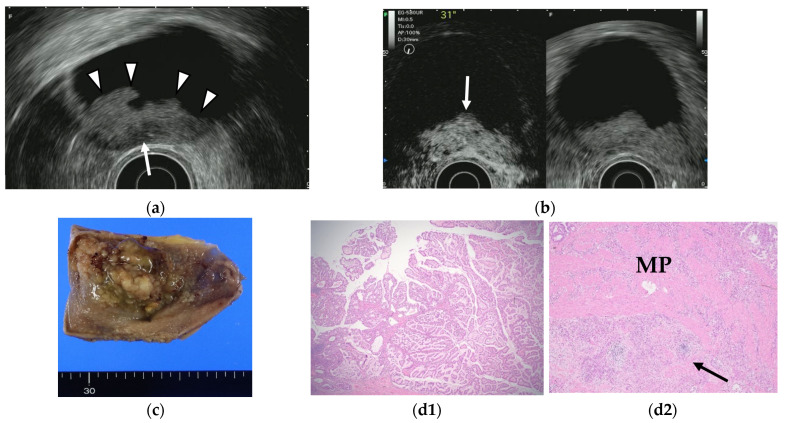
GB carcinoma. (**a**) Conventional EUS shows elevated lesions with conspicuous surface irregularities (arrowheads) observed in the gallbladder body. A hypoechoic region is observed in the deep part of the lesion (arrow), and the outer hyperechoic layer is also irregular, suggesting infiltration into the subserosal layer. (**b**) The contrast-enhanced harmonic EUS image after the injection of Sonazoid^®^ shows that lesions in the gallbladder body (arrow) have a strong heterogeneous staining effect from the early stage of contrast enhancement. (Left contrast-enhanced harmonic mode, right B-mode) (**c**) Photograph of the gross pathologic specimen after cholecystectomy shows that the papillary neoplasm with a maximum diameter of 55 mm is found from the body to the bottom of the gallbladder. (**d1**,**d2**): H-E stain of the specimen demonstrates atypical epithelial cells grow papillary. Infiltration into the subserosal layer is observed in a part of the deep part of the tumor with infiltration and hyperplasia of poorly differentiated adenocarcinoma. (MP muscularis propria).

**Figure 8 diagnostics-11-01789-f008:**
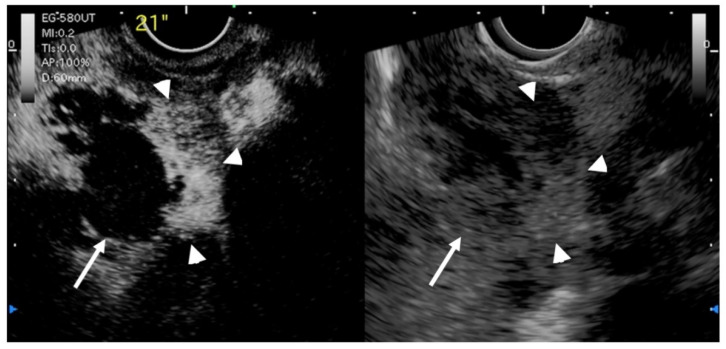
EUS image of GB carcinoma. Irregular wall-thickening of the GB (arrowhead) is observed. In the conventional EUS image (**right**), a structure is found inside the GB and the lumen is unknown. The contrast-enhanced harmonic image 21 s after the injection of Sonazoid^®^ (**left**) shows heterogeneous enhancement in the thickened wall (arrowhead). The structure inside the GB is not enhanced and can be diagnosed as biliary sludge (arrow) rather than a neoplasm.

**Figure 9 diagnostics-11-01789-f009:**
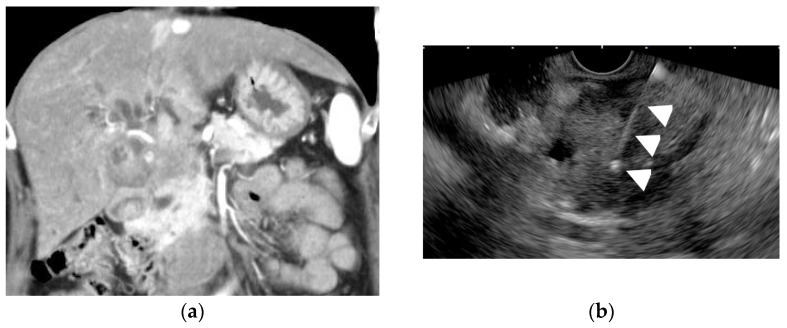
Endoscopic ultrasound-guided fine-needle aspiration (EUS-FNA) for a GB lesion. (**a**) Enhanced CT scan shows a GB lesion. (**b**) EUS-FNA of a GB mass lesion. The arrowhead shows the FNA needle inside the lesion. The pathology result showed adenocarcinoma.

**Table 1 diagnostics-11-01789-t001:** Classification of GB protuberant lesions.

Protuberant lesions	Neoplastic	Adenoma	Carcinoma
	non-neoplastic	cholesterol polyp	hyperplastic polyp
		inflammatory polyp	fibrous polyp
		metaplastic polyp	adednomyomatosis

**Table 2 diagnostics-11-01789-t002:** Classification of GB wall-thickening lesions.

GB wall-thickening lesions	Neoplastic	Cacinoma	Lymphoma
	non-neoplastic	inflammation	acute cholecystitis
			chronic cholecystitis
			xanthogranulomatous cholecystitis
		hyperplasia	adenomyomatosis
			hyperplasia accompanying anomalous pancreaticobiliary junction

**Table 3 diagnostics-11-01789-t003:** EUS features of major GB protuberant lesions.

	Size	Pedunculation	Morpholigy	Surface	Internal Echo
**Cholesterol polyp**	<10 mm	pedunculated	morular	deeply notched granular	rough or granular hyperechoic spots
**Hyperplastic polyp**	≥10 mm	pedunculated	papillated or lobulated	smooth	uniform low echogenicity
**Adenomyomatosis**	no fixed size	sessile	oval	relatively smooth or granular	multiple anechoic aresa comet tail artifact
**Adenoma**	5–20 mm	pedunculated or subpedunculated	oval	nodular or relatively smooth	homogeneously isoechoic multiple microcystic spaces
**Carcinoma**	≥10 mm	sessile > pedunculated	oval or irregular	nodular or smooth	heterogeneously dense echogenic hypoechoic areas at the cores

**Table 4 diagnostics-11-01789-t004:** EUS features of major GB wall-thickening lesions.

	Extent	Surface Structure of Lumen	Internal Stricture	Layer Structure
**Acute Cholecystitis**	diffuse	smooth	no distinctive findings	preserved Sonolucent layer, striations
**Adenomyomatosis**	focal or diffuse	smooth	multiple anechoic areas comet tail artifact	preserved
**Xanthogranulomatous cholecystitis**	focal or diffuse	smooth	mixed hyperechoic and hypoechoic echotexture	irregular or disrupted
**Hyperplasia accompanying anomalous pancreaticobiliary junction**	diffuse	smooth	uniform hypoechogenicity	preserved
**Carcinoma**	focal > diffuse thickness > 10 mm	Irregular or papillated	uneven hypoechogenicity	irregular or disrupted (in advance lesions)

**Table 5 diagnostics-11-01789-t005:** EUS classification of GB carcinoma and correlation with T staging (adapted from [49]).

EUS Classification Type	Shape	Surface	Outer Hyperechoic Layer	T Staging
A	Pedunculated	Nodular	Intact	Tis (-1)
B	broad-based protrusion or wall-thickening	irregular	intact	T1–2
C	broad-based protrusion or wall-thickening	irregular	irregular	T2
D	broad-based protrusion or wall-thickening	irregular	disrupted	T3–4

**Table 6 diagnostics-11-01789-t006:** The sensitivity and specificity on CH-EUS for the diagnosis of GB malignancy.

Author	Year	Study Design	Patients	Contrast Agent	Sensitivity	Specificity
Hirooka [50]	1998	retrospective	38	Albunex	0.79	0.54
Choi [51]	2013	retrospective	90	SonoVue	0.94	0.93
Imazu [53]	2014	retrospective	36	Sonazoid	0.90	0.98
Sugimoto [54]	2016	retrospective	24	Sonazoid	1.00	0.94
Kamata [52]	2017	retrospective	125	Sonazoid	0.90	0.98
Leem [55]	2018	retrospective	145	SonoVue	0.97	0.55
Liang, X [56]	2020	meta-analysis	458		0.92	0.89

**Table 7 diagnostics-11-01789-t007:** The results on EUS-FNA for the diagnosis of GB malignancy.

Author	Year	Patients	Sensitivity	Specificity	Accuracy	Complication
Jacobson [59]	2003	6	0.80	1.00	0.83	None
Varadarajulu [60]	2005	6	1.00	1.00	1.00	None
Meara [61]	2006	7	0.80	1.00	0.86	None
Hijioka [42]	2010	15	0.90	1.00	0.93	None
Kim [62]	2012	21	0.93			Cholecystitis
Hijioka [58]	2012	50	0.96	1.00	0.98	None
Ogura [63]	2014	16	1.00	1.00	1.00	None
Singla [64]	2018	101	0.91	1.00	0.91	none

## Data Availability

Data supporting reported results can be found in the references at the end of this article.

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
