# Peer review of "The Role of Endoscopic Ultrasound in the Diagnosis of Gallbladder Lesions"

_diagnostics, 2021, doi:10.3390/diagnostics11101789_

Round 1

Reviewer 1 Report

Dear Hashimoto and colleagues,

I have had an opportunity to review your manuscript. This has been written appropriately and up-to-date.

The manuscript is well-written and up-to-date and covers the most important areas ( gallstones, cholesterol polyps, adenomyomatosis, and GB carcinoma, differential diagnosis of GB lesions).

Best wishes.

Author Response

Thank you for your reviewing our manuscript.

Reviewer 2 Report

Dear Editor, this article entitled “The role of endoscopic ultrasound in the diagnosis of gallbladder diseases” aims to review the role of EUS in the diagnosis of gallbladder (GB) pathologies.

I have the following comments:

-In consideration of the “review” nature of the article, it should be better if the Authors briefly specify their search strategy and how this was conducted i.e., through the most powerful research engines, such as MEDLINE, EMBASE, PubMed, Scopus, Web of Science, Cochrane Library?  

-The Authors attempted to report all data providing a comprehensive assessment of the reported studies evaluating the diagnostic accuracy using EUS in diagnosing all BG pathologies. The subject of GB diseases seems too large considering that more than 90% of GB diseases are gallstone-related. it should be better to consider restricting the subject to GB lesions and not to GB diseases, where the role of EUS can be less important and only in a particular clinical setting (idiopathic acute pancreatitis).

-Tables summarizing principle EUS features for different GB lesions can be very handy for readers. There is no in the paper clear summary of EUS features with the most important criteria to achieve a correct differential diagnosis of all GB lesions. Furtermore, reporting in a Table the range of accuracy for EUS can be a useful effort.

-Some Authors routinely divide gallbladder lesions simply into two broad families 1) protuberant lesions and 2) wall-thickening lesions. Probably by simplifying a classification of the lesions can be helpful to clarify some clinical concepts.  The protuberant lesion is a generic term for lesions that have the specific morphological feature of forming a protuberance localized to the luminal side of the gallbladder, which encompasses a variety of diseases, both epithelial and non-epithelial, as well as benign and malignant diseases. However, it should be also helpful from a clinical perspective of treating lesions to correctly define all protuberant gallbladder lesions into neoplastic and non-neoplastic lesions and to describe for readers in a specific Table what can be the most important EUS  features and their close range of accuracy for differentiating adenomas from carcinomas or other lesions such as cholesterol polyps, hyperplastic polyps, and gallbladder adenomyomatosis, xanthogranulomatous chlolecystitis, based on their size, pedunculation, morphology, surface characteristics, and internal echo findings.

-Wall-thickening lesions denote lesions in which the gallbladder wall is diffusely thickened. The differential diagnosis for these lesions should also be described with reference to the extent of wall thickening, surface structure, and presence or absence of Rokitansky–Aschoff sinuses (RAS).

-Please try to give some clear indications for fine needle-EUS biopsy and give some indications about a correct technique above all in those patients with a suspected diagnosis of neoplastic invasion of the liver parenchyma.

Round 2

Reviewer 2 Report

I aknowledge the efforts that the Authors have done in revising their manuscript . In my opinion the Authors have satisfactorily pointed out and answered all the questions and critical points underlined by the reviewer, adding some useful tables which is really a good job. I have no further comments on the paper and I consider the manuscript an interesting and well written narrative review.